# Inactivation of the type I interferon pathway reveals long double-stranded RNA-mediated RNA interference in mammalian cells

Pierre V Maillard[1,**], Annemarthe G Van der Veen[1], Safia Deddouche-Grass[1,†], Neil C Rogers[1], Andres Merits[2] & Caetano Reis e Sousa[1,*]

## Abstract

RNA interference (RNAi) elicited by long double-stranded (ds) or base-paired viral RNA constitutes the major mechanism of antiviral defence in plants and invertebrates. In contrast, it is controversial whether it acts in chordates. Rather, in vertebrates, viral RNAs induce a distinct defence system known as the interferon (IFN) response. Here, we tested the possibility that the IFN response masks or inhibits antiviral RNAi in mammalian cells. Consistent with that notion, we find that sequence-specific gene silencing can be triggered by long dsRNAs in differentiated mouse cells rendered deficient in components of the IFN pathway. This unveiled response is dependent on the canonical RNAi machinery and is lost upon treatment of IFN-responsive cells with type I IFN. Notably, transfection with long dsRNA specifically vaccinates IFN-deficient cells against infection with viruses bearing a homologous sequence. Thus, our data reveal that RNAi constitutes an ancient antiviral strategy conserved from plants to mammals that precedes but has not been superseded by vertebrate evolution of the IFN system.

**Keywords** RNA interference; double-stranded RNA; innate immunity; viral infection
**Subject Categories** Immunology; RNA Biology
**The EMBO Journal (2016) 35: 2505–2518**

See also: **JM Luna et al** (December 2016)

## Introduction

RNA silencing is a mechanism conserved from plants to mammals that mediates suppression of gene expression through the use of small RNAs of 21–30 nucleotides (nt) in length (Ghildiyal & Zamore, 2009; Wilson & Doudna, 2013). These small RNAs guide a wide range of activities such as the regulation of gene expression, establishment of heterochromatin, control of transposons and defence against virus infection (Ghildiyal & Zamore, 2009; Wilson & Doudna, 2013). Most—albeit not all—mechanisms of RNA silencing operate through a similar set of processes that are triggered by the presence within cells of double-stranded RNA (dsRNA) or single-stranded RNA (ssRNA) containing long base-paired regions. This was initially demonstrated in *C. elegans*, in which introduction of long dsRNA induced the selective degradation of mRNAs bearing complementary sequences (Fire *et al*, 1998). The process was termed RNA interference (RNAi) and relies on the processing of a perfectly base-paired dsRNA by the type III ribonuclease Dicer into a class of small RNAs termed small interfering RNAs (siRNAs). One strand of siRNA is loaded onto Argonaute (Ago) family proteins to form the RNA-induced silencing complex (RISC), which is then directed to recognise and cleave cellular mRNAs bearing complementary sequences (Meister & Tuschl, 2004; Ghildiyal & Zamore, 2009; Wilson & Doudna, 2013). By its very nature, RNAi constitutes a sequence-specific gene silencing mechanism.

Virus infection is often accompanied by generation of dsRNA or long base-paired RNAs (Jacobs & Langland, 1996; Weber *et al*, 2006; Son *et al*, 2015). These can serve as substrates for Dicer to generate virus-derived siRNAs (viRNAs) (Ding & Voinnet, 2007; Kemp & Imler, 2009; Swarts *et al*, 2014; tenOever, 2016) that are loaded onto RISC and target complementary viral RNA to block virus replication. Indeed, viRNA-mediated RNAi constitutes the primary antiviral defence strategy of plants and invertebrates (Ding & Voinnet, 2007; Kemp & Imler, 2009; Swarts *et al*, 2014; tenOever, 2016). Consequently, plant and insect viruses have evolved virulence factors called viral suppressors of RNA silencing (VSRs) that block various steps of the antiviral RNAi mechanism (Haasnoot *et al*, 2007; Wu *et al*, 2010). Experimental appreciation of the effects of RNAi on virus accumulation therefore requires the use of VSR-deficient viruses (Deleris *et al*, 2006; Wang *et al*, 2006a; Diaz-Pendon *et al*, 2007; Maillard *et al*, 2013).

1  Immunobiology Laboratory, The Francis Crick Institute, London, UK
2  Institute of Technology, University of Tartu, Tartu, Estonia
   *Corresponding author. Tel: +44 20 3796 1310; E-mail: caetano@crick.ac.uk
   **Corresponding author. Tel: +44 20 3796 1280; E-mail: pierre.maillard@crick.ac.uk
   †Present address: Open Innovation Access Platform, Sanofi Strasbourg, Strasbourg, France

    The EMBO Journal   Vol 35 | No 23 | 2016    **2505**

Whether RNAi also acts as an antiviral mechanism in mammals is hotly debated (Parameswaran *et al*, 2010; Cullen *et al*, 2013; Li *et al*, 2013; Maillard *et al*, 2013; Backes *et al*, 2014; Ding & Voinnet, 2014; Kennedy *et al*, 2015). Infection of mammalian somatic cells with various viruses results in little accumulation of viRNA (Parameswaran *et al*, 2010; Girardi *et al*, 2013; Backes *et al*, 2014; Bogerd *et al*, 2014) and, where tested, virus replication is only modestly affected in Dicer-defective cells (Wang *et al*, 2006b; Matskevich & Moelling, 2007; Bogerd *et al*, 2014). Instead, infection of vertebrate cells with viruses triggers the innate interferon (IFN) response, a vertebrate-restricted system of antiviral defence that can be elicited by the same ds and base-paired RNAs that serve as Dicer substrates. In the IFN response, those RNAs are generally detected by members of the RIG-I family of pattern recognition receptors, which then bind to the mitochondrial adaptor MAVS to initiate a signalling cascade that culminates in the production and secretion of IFN-β and IFN-α, among others (Goubau *et al*, 2013). IFN-α and IFN-β bind to the IFN receptor (IFNAR), which signals to induce hundreds of interferon-stimulated genes (ISGs) that work to increase antiviral resistance (Samuel, 2001; Goubau *et al*, 2013; Schneider *et al*, 2014). For example, the ISG protein kinase R (PKR), when activated by viral dsRNA, phosphorylates the alpha subunit of the protein synthesis initiation factor-2, causing inhibition of both viral and cellular translation (Pindel & Sadler, 2011). Another ISG, the enzyme 2′,5′-oligoadenylate synthetase (2′-5′ OAS), is similarly activated by viral dsRNA to produce oligo-adenylates that, in turn, induce activation of another ISG, RNase L, which non-specifically degrades RNA of viral or cellular origins (Chakrabarti *et al*, 2011). Thus, introduction of viral RNA or its mimics in mammalian cells does not results in sequence-specific gene silencing, but rather in a global sequence-unspecific reduction in protein expression and often culminates in cell death (Yang *et al*, 2001; Paddison *et al*, 2002). As such, a consensus has emerged that long dsRNA-mediated RNAi (dsRNAi) and viral dsRNA-mediated RNAi (antiviral RNAi) have been lost during the course of vertebrate evolution to be replaced by the IFN system of antiviral defence (Cullen *et al*, 2013; tenOever, 2016).

Despite the above observations, long dsRNA has been shown to induce RNAi in some undifferentiated mammalian cell types, such as mouse embryonic stem cells (mESCs), mouse embryonic terato-carcinoma cell lines and in oocytes (Svoboda *et al*, 2000; Wianny & Zernicka-Goetz, 2000; Billy *et al*, 2001; Yang *et al*, 2001; Paddison *et al*, 2002). For example, viral infection of mESCs resulted in RISC loading with viRNAs that could provide antiviral activity towards a VSR-deficient homologous virus (Maillard *et al*, 2013). Interestingly, oocytes express an alternative isoform of Dicer that lacks the N-terminal helicase domain (Flemr *et al*, 2013) and has enhanced ability to process long dsRNA. Overexpression of the N-terminal truncated form of human Dicer in 293T cells lacking endogenous Dicer and PKR conferred the ability to produce functional amounts of siRNAs and, to some extent, viRNAs (Kennedy *et al*, 2015). These observations could indicate that failure to express an appropriate Dicer isoform underlies the inability of differentiated cells to carry out dsRNAi and antiviral RNAi. A distinct but not mutually exclusive possibility is that the presence of the IFN response, which is attenuated in undifferentiated and germ cells (Wood & Hovanessian, 1979; Stein *et al*, 2005; Wang *et al*, 2013), may mask or inhibit dsRNAi and antiviral RNAi in differentiated mammalian cells. Here, we provide evidence for the latter notion by showing that inactivation of the IFN pathway in mammalian somatic cells unveils the presence of long dsRNA-mediated RNAi. Our data suggest that long dsRNAi and antiviral RNAi could constitute a backup strategy of antiviral defence that has been preserved during the evolution of vertebrate immunity.

# Results

The hallmark of RNAi is specific silencing of a target RNA upon introduction of long dsRNA bearing a complementary sequence (Meister & Tuschl, 2004; Ghildiyal & Zamore, 2009; Wilson & Doudna, 2013). As mentioned previously, failure to document this process in the context of viral infection is complicated by the possibility that the virus encodes VSRs (Haasnoot *et al*, 2007; Wu *et al*, 2010). Therefore, we designed a virus-free assay to measure long dsRNA-mediated gene silencing at the single-cell level and tested it initially in mouse embryonic stem cells (mESCs), in which the presence of dsRNAi and antiviral RNAi has been reported (Billy *et al*, 2001; Yang *et al*, 2001; Paddison *et al*, 2002; Maillard *et al*, 2013). mESCs stably expressing a GFP reporter gene were transfected with *in vitro* synthesised Cy5-labelled long dsRNA corresponding to the first 200 nt of either the GFP (dsRNA-GFP) or, as a control, the *Renilla* luciferase (dsRNA-*RL*) coding sequences; GFP expression in live single Cy5-positive (transfected) cells was then measured by flow cytometry (Fig 1A and B). Notably, introduction of dsRNA-*GFP* but not dsRNA-*RL* led to a reduction in GFP expression at 2 days post-transfection (Fig 1C). To increase temporal resolution, we switched to mESCs expressing a destabilised form of GFP (d2GFP), which has a reduced half-life (Li *et al*, 1998). At 6 h post-transfection (h.p.t.), d2GFP expression was decreased by either of the dsRNAs, reflecting sequence-unspecific effects (Fig 1D). However, at later time points (24–48 h.p.t.), a reduction in d2GFP signal was maintained and accentuated in mESCs transfected with dsRNA-*GFP* but not in cells treated with dsRNA-*RL*, revealing sequence-specific gene silencing (Fig 1D). In contrast, transfection of the same long dsRNAs into a mouse fibroblast cell line (L929) stably expressing d2GFP resulted in a sequence-unspecific decrease in GFP signal at all time points examined (Fig 1E). We also noticed that transfection of long dsRNA induced some cytotoxicity in mESCs and L929 cells (Appendix Fig S1A and B). This is consistent with the notion that introduction of long dsRNA mimics virus infection and causes IFN-dependent cellular shutdown/death, even in mESCs that have functional PKR despite an attenuated IFN pathway (Wang *et al*, 2013).

To address whether the IFN response in differentiated cells was connected to the failure to reveal sequence-specific gene silencing, we generated immortalised GFP-expressing *Mavs*$^{-/-}$ mouse embryonic fibroblasts (MEFs) (Appendix Fig S2A). *Mavs*$^{-/-}$ MEFs did not mount an IFN response to dsRNA, unlike their MAVS-sufficient (*Mavs*$^{+/-}$) counterparts, which reacted to dsRNA but not siRNA with upregulation of ISGs and decreased viability (Appendix Fig S2B and C). GFP expression in viable control *Mavs*$^{+/-}$ MEFs was unaltered by treatment with either dsRNA-*RL* or dsRNA-*GFP*, indicating the lack of gene silencing induced by long dsRNA, but could be reduced by treatment with siRNAs targeting GFP confirming the presence of a functional RISC (Fig 2A). In contrast, in *Mavs*$^{-/-}$ MEFs, GFP levels were specifically reduced by dsRNA-*GFP*, as well as siRNA targeting GFP, but not by dsRNA-*RL* (Fig 2A). Importantly, silencing induced by dsRNA-*GFP* was not restricted to a subset of

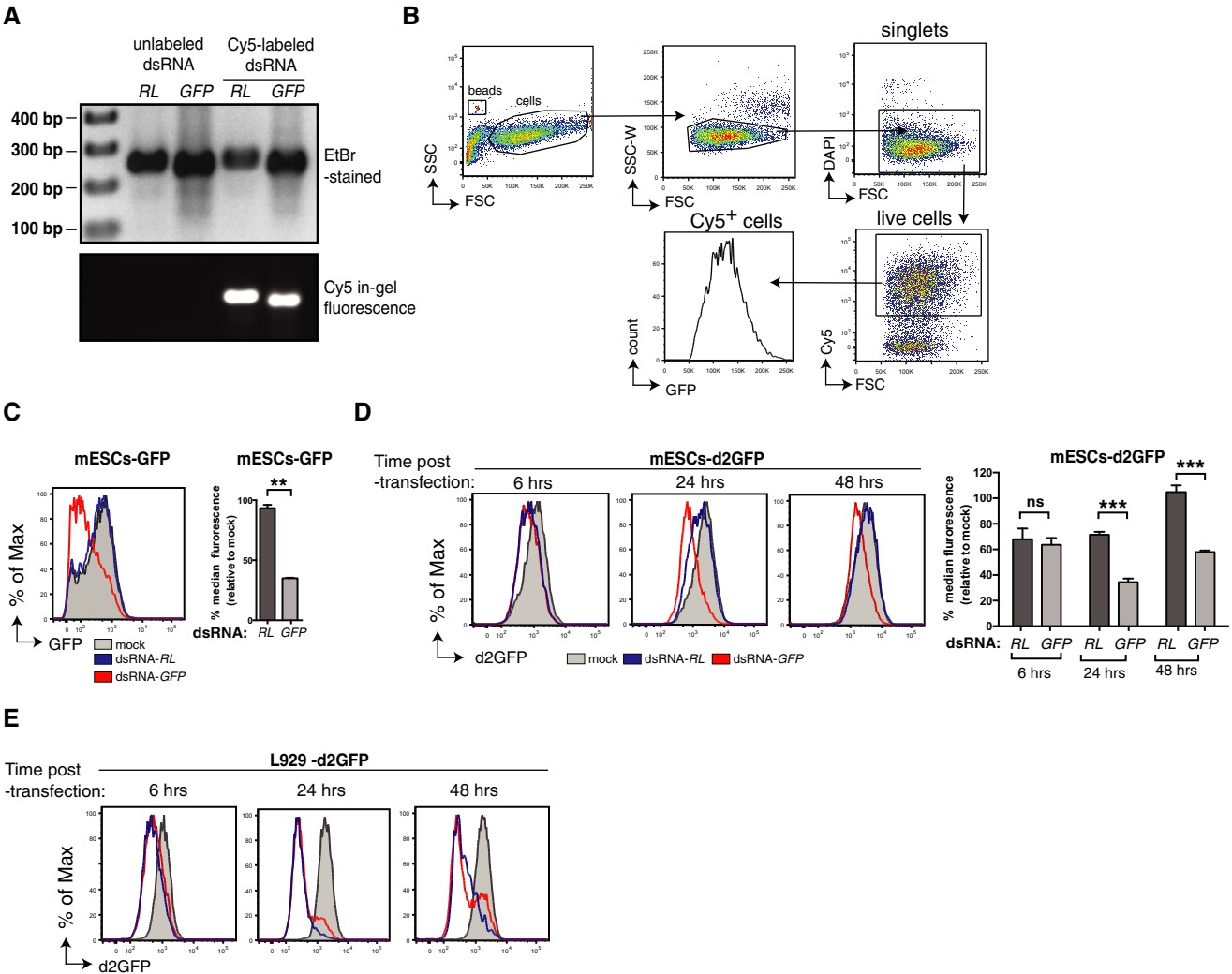

**Figure 1. Flow cytometry-based assay to monitor gene silencing by long dsRNA.**

A   Unlabelled or Cy5-labelled dsRNA corresponding to the first 200 nucleotides of either the *Renilla* luciferase (dsRNA-*RL*) or the GFP (dsRNA-*GFP*) coding sequences visualised on gel stained with ethidium bromide (EtBr-stained, top) or by in-gel fluorescence (bottom).

B   Gating strategy used in the flow cytometry-based assay. First, cells were selected (side scatter (SSC) vs. forward scatter (FSC)) and gated for singlets (SSC-Width (W) vs. FSC) to hence exclude doublets or cell aggregates. The single-cell gate was then further analysed for DAPI, gating on the live cells (DAPI negative). DsRNA-transfected cells (Cy5 positive) were then gated, and the level of GFP expression was analysed within that population. Flow cytometry counting beads (gated on SSC vs. FSC plot) were added to quantify cell number in each sample.

C   mESCs stably expressing GFP (mESCs-GFP) were mock transfected or transfected with the indicated Cy5-labelled dsRNA, and the level of GFP in Cy5+ cells was analysed 3 days later and is represented as histograms (left). Right panel, bar graphs displaying the percentage of GFP median fluorescence intensity relative to mock control.

D   mESCs stably expressing d2GFP (mESCs-d2GFP) were mock transfected or transfected with the indicated Cy5-labelled dsRNA. The level of d2GFP was analysed at the indicated time points post-transfection and displayed as in (C).

E   L929 cells stably expressing d2GFP were mock transfected or transfected with the indicated Cy5-labelled dsRNA, and d2GFP level was measured as in (D).

Data information: All histogram plots are representative of two independent experiments. Each histogram and bar represents a sample size of 10,000 cells. Bar graphs depict the median fluorescence values normalised to those in mock-transfected samples. Mean values and standard deviations (SD) from two independent experiments are shown (C, D). Statistical analysis was performed using two-way ANOVA with Sidak's multiple comparisons test as post-test for unpaired *t*-test (C) or pairwise comparisons (D). Significant differences with Student's unpaired *t*-test or Sidak's multiple comparisons test are shown (ns, not significant; **$P < 0.01$; ***$P < 0.001$). Source data are available online for this figure.

putatively undifferentiated *Mavs*$^{-/-}$ MEFs as the entire transfected population shifted towards lower levels of expression (Fig 2A). The sequence-specific knock-down of GFP was also apparent at the mRNA level in *Mavs*$^{-/-}$ MEFs and, to a lesser extent, in *Mavs*$^{+/-}$ heterozygous MEFs (Appendix Fig S2D).

We repeated the experiments in *Mavs*$^{+/-}$ and *Mavs*$^{-/-}$ MEFs expressing d2GFP to obtain temporal resolution of dsRNA-induced gene silencing. At 6 h.p.t., d2GFP levels were reduced by both dsRNA-*RL* and dsRNA-*GFP* in MEFs irrespective of genotype (Fig 2B). The sequence specificity of the response only became

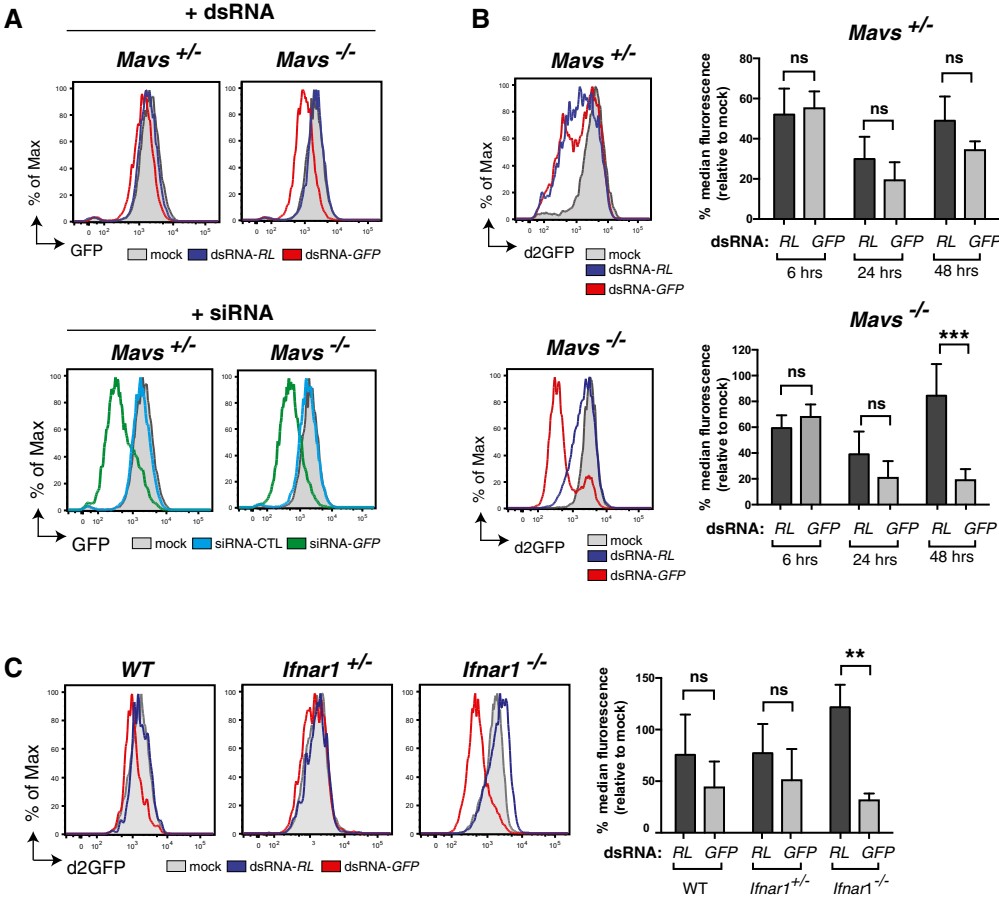

**Figure 2. Sequence-specific gene silencing by long dsRNA in cells deficient in the IFN response.**

A   *Mavs*[+/−] and *Mavs*[−/−] MEFs stably expressing GFP were transfected or not (mock) with the indicated Cy5-labelled dsRNAs (top panels) or siRNAs (bottom panels). GFP level was measured in Cy5[+] cells by flow cytometry 72 h (hours) post-transfection.

B   *Mavs*[+/−] and *Mavs*[−/−] MEFs stably expressing d2GFP were transfected or not (mock) with the indicated Cy5-labelled dsRNAs. The level of d2GFP in Cy5[+] cells was measured at various time points post-transfection as indicated. Histogram plots show d2GFP level 48 h post-transfection. Bar graphs show three time points as indicated.

C   WT, *Ifnar1*[+/−] or *Ifnar1*[−/−] MEFs stably expressing d2GFP were transfected or not (mock) with the indicated Cy5-labelled dsRNAs, and d2GFP level in Cy5[+] cells was monitored by flow cytometry at 48 h post-transfection.

Data information: All histogram plots are representative of 10 (A, B) or 4 (C) independent experiments. Each histogram and bar represents a sample size of 10,000 cells. Bar graphs depict the median fluorescence values normalised to those in mock-transfected samples. Mean values and SD of three independent experiments are shown (B, C). Statistical analysis was performed using two-way ANOVA with Sidak's multiple comparisons test as post-test for pairwise comparisons. Significant differences with Sidak's multiple comparisons test are shown (ns, not significant; **$P < 0.01$; ***$P < 0.001$).

apparent in *Mavs*[−/−] MEFs at 24–48 h.p.t. with dsRNA-*GFP* (Fig 2B). Sequence-specific dsRNA-dependent gene silencing was also observed in MEFs derived from type I IFN receptor-deficient mice (*Ifnar1*[−/−] MEFs) that are competent for type I IFN production but defective in ISG induction (Fig 2C). Similarly, we could unveil sequence-specific dsRNA-dependent gene silencing in *Mavs*[+/−] MEFs by blocking IFNAR with a neutralising antibody (Fig 3A). In contrast, blocking IFNAR in *Mavs*[−/−] MEFs did not impact the sequence-specific response, consistent with the fact that they do not produce IFN in response to dsRNA (Fig 3A). The converse experiment, stimulation of IFNAR in *Mavs*[−/−] MEFs by addition of recombinant IFN, caused a sequence-unspecific decrease in d2GFP expression in response to either dsRNA-*GFP* or dsRNA-*RL* (Fig 3B), while allowing sequence-specific knock-down of d2GFP by siRNAs

(Appendix Fig S3). In summary, the IFN response to dsRNA in differentiated mammalian cells induces ISGs that mask or suppress sequence-specific responses to long dsRNA.

We next addressed whether the unveiled sequence-specific gene silencing mediated by long dsRNA in cells deficient in the IFN response is mediated by the canonical RNAi pathway. A hallmark of RNAi is the production of 21–25-nt siRNAs following processing by Dicer of long dsRNA. We therefore transfected *Ifnar1*[−/−] MEFs with dsRNA-*RL* or dsRNA-*GFP* and analysed total RNA by Northern blot one day later using appropriate probes. In both sets of transfected cells, we detected sequence-specific RNAs of ~22 nt in length, in line with the size of siRNAs typically generated by mammalian Dicer (Zhang *et al*, 2002; Vermeulen *et al*, 2005; Flemr *et al*, 2013; Maillard *et al*, 2013; Kennedy *et al*, 2015) and similar to the size

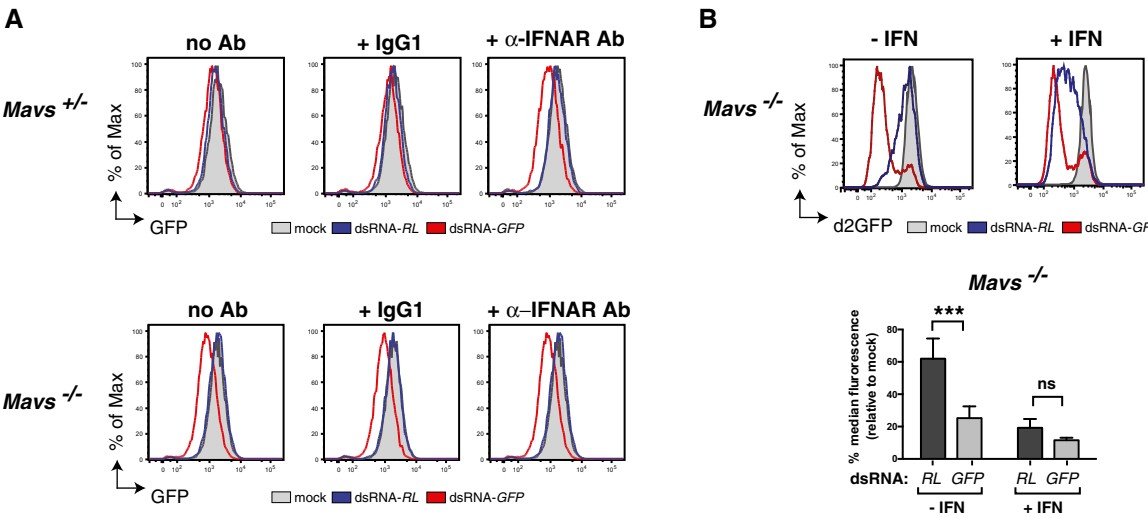

**Figure 3.  The IFN pathway negatively influences the detection of the sequence-specific gene silencing.**

A   *Mavs*[+/−] and *Mavs*[−/−] MEFs stably expressing GFP were incubated with either a neutralising antibody against IFNAR (α-IFNAR Ab), an isotype control antibody (IgG1) or no antibody (no Ab). Cells were then transfected or not (mock) with Cy5-labelled dsRNA as indicated and GFP level monitored in Cy5[+] cells 72 h post-transfection.

B   *Mavs*[−/−] MEFs stably expressing d2GFP were treated with recombinant IFN (200 U ml[−1]) for 24 h prior to transfection or not (mock) with Cy5-labelled dsRNA, as indicated. IFN treatment was maintained for 2 days post-transfection and then the GFP level was monitored in Cy5[+] cells.

Data information: All histogram plots are representative of two (A) or three (B) independent experiments. Each histogram and bar represents a sample size of 10,000 cells. Bar graphs depict median fluorescence values normalised to those in mock-transfected samples. Mean values and SD of three independent experiments are shown (B). Statistical analysis was performed using two-way ANOVA with Sidak's multiple comparisons test as post-test for pairwise comparisons. Significant differences with Sidak's multiple comparisons test are shown (ns, not significant; ***$P < 0.001$).

generated by incubating dsRNA-*RL* or dsRNA-*GFP* with human Dicer complexes *in vitro* (Fig 4A). We then directly addressed the role of Dicer in dsRNA-transfected cells. To obviate the detrimental effect of complete Dicer ablation, which would prevent miRNA generation and lead to strong selection for atypical clones (Kim *et al*, 2009), we chose to decrease its expression in *Mavs*[−/−] MEFs expressing d2GFP by using shRNA-mediated silencing (Fig 4B). The sequence-specific

silencing of d2GFP by dsRNA-*GFP* was highly reduced following knock-down of Dicer (*DICER* shRNA #1) but was unaffected by either a non-targeting shRNA control or an inefficient shRNA against Dicer (*DICER* shRNA #2) (Fig 4C). Next, we evaluated the role of RISC. Of the four mammalian Ago proteins (Ago1–4), only Ago2 is essential for experimental RNAi induced by siRNA (Meister & Tuschl, 2004; Wilson & Doudna, 2013). We therefore used CRISPR

**Figure 4.  Sequence-specific gene silencing by long dsRNA depends on Dicer and Ago2.**

A   Northern blot analysis of *Ifnar1*[−/−] MEFs at 24 h.p.t. with the indicated dsRNA or transfection reagent alone (mock) using a probe specific for dsRNA-*GFP* (top) or dsRNA-*RL* (bottom). Reactions from a dicing assay using the indicated dsRNA incubated *in vitro* with immunoprecipitated FLAG-human Dicer were loaded in parallel. Two lanes (delimited by dashed lines) were left empty between the *Ifnar1*[−/−] MEFs transfected samples and the *in vitro* dicing assay reactions. A miRNA marker containing three synthetic ssRNA oligonucleotides of 17, 21 and 25 nt in length was run in parallel. Endogenous U6 was used as a loading control. Signals corresponding to the dsRNA-derived siRNAs are indicated with an arrow. The membrane was first probed for dsRNA-*GFP* (top), then probed for the miRNA marker (top left) and subsequently stripped and reprobed for dsRNA-*RL* (middle) and finally stripped and reprobed for both U6 (bottom) and for the miRNA marker again (middle left).

B   Western blot of Dicer in d2GFP-expressing *Mavs*[−/−] MEFs either not transduced (NT) or transduced to express a non-targeting shRNA control (CTL shRNA) or two different Dicer-specific targeting shRNAs (*DICER* shRNA #1 and #2). P97 served as a loading control.

C   Cells described in (B) were transfected or not (mock) with the indicated Cy5-labelled dsRNAs, and d2GFP level in Cy5[+] cells was monitored by flow cytometry 48 h later. Please note that the biphasic pattern of d2GFP level observed in dsRNA-*GFP* transfected cells is caused by a lower transfection efficiency of dsRNA in these cells giving rise to significant amounts of lowly transfected cells that do not display a decreased level of d2GFP.

D   Western blots of Ago2 in parental *Mavs*[−/−] MEFs stably expressing d2GFP and three *Mavs*[−/−] *Ago2*[−/−] clones generated by CRISPR/CAS9-mediated genome engineering. P97 served as a loading control.

E   *Mavs*[−/−] (parental line) and *Mavs*[−/−] *Ago2*[−/−] MEFs (clone 5.1) stably expressing d2GFP were transfected or not (mock) with the indicated Cy5-labelled dsRNAs and d2GFP level in Cy5[+] cells was monitored by flow cytometry 48 h later.

Data information: Histogram plots are representative of two independent experiments (C) or of two independent CRISPR clones (E). Each histogram and bar represents a sample size of 10,000 cells. Bar graphs depict median fluorescence values normalised to those in mock-transfected samples. Mean values and SD of two independent experiments (C) or of two independent CRISPR clones (B) are shown. Statistical analysis was performed using two-way ANOVA with Sidak's multiple comparisons test as post-test for pairwise comparisons. Significant differences with Sidak's multiple comparisons test are shown (ns, not significant; *$P < 0.05$; ***$P < 0.001$).
Source data are available online for this figure.

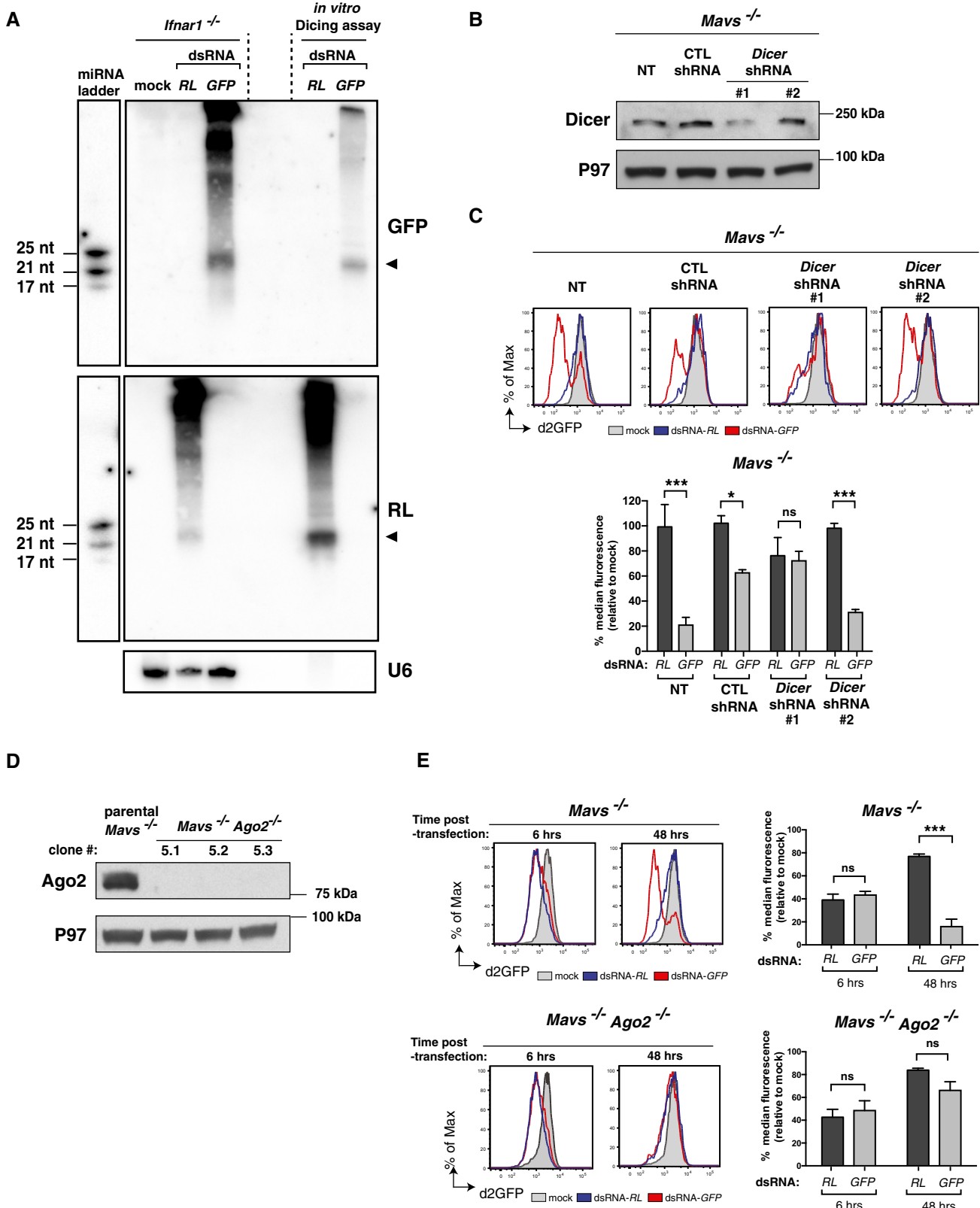

Figure 4.

technology to delete *Ago2* in *Mavs*$^{-/-}$ MEFs expressing d2GFP (Fig 4D). The early sequence-unspecific decrease in d2GFP levels induced by either dsRNA-*RL* or dsRNA-*GFP* was not affected by loss of Ago2 in CRISPR-targeted clones (Fig 4E). In contrast, the sequence-specific silencing of d2GFP observed 48 h.p.t. of dsRNA-*GFP* in *Mavs*$^{-/-}$ MEFs was completely abolished in *Mavs*$^{-/-}$ *Ago2*$^{-/-}$ MEFs (Fig 4E). We independently generated different *Mavs*$^{-/-}$ *Ago2*$^{-/-}$ MEFs clones expressing stable GFP rather than d2GFP and confirmed the essential role of Ago2 in the sequence-specific response to long dsRNAs, as well as to siRNAs (Appendix Fig S4A–C). We further generated *Ago1*$^{-/-}$-CRISPR-targeted clones and showed that, in contrast to Ago2, Ago1 is dispensable for dsRNA-dependent sequence-specific gene silencing (Appendix Fig S4D–F).

Ago2 is the only member of the Ago family capable of catalysing endonucleolytic cleavage ("slicing") of target transcripts, an activity that has been conserved in vertebrates (Tolia & Joshua-Tor, 2007). To assess the importance of the "slicing" activity of Ago2 in the sequence-specific response to long dsRNA, we complemented *Ago2*$^{-/-}$ *Mavs*$^{-/-}$ clones with a wild-type (WT) or a catalytic mutant (D597A) version of mouse Ago2 (mAgo2; Fig 5A). As previously reported (Liu *et al*, 2004; Meister *et al*, 2004), WT mAgo2 but not mutant mAgo2 D597A supported silencing of GFP induced by siRNA (Appendix Fig S5). Similarly, only the WT but not the catalytic mutant version of mAgo2 was able to restore sequence-specific GFP silencing in *Ago2*$^{-/-}$ *Mavs*$^{-/-}$ clones transfected with dsRNA-*GFP* (Fig 5B). We conclude that the sequence-specific response to long dsRNA in IFN-deficient cells critically depends on the catalytic activity of Ago2, as well as on Dicer, and, therefore, fulfils the criteria of long dsRNA-mediated RNAi (dsRNAi).

We then addressed whether dsRNAi can be used as a mechanism for antiviral defence in mammalian cells independently of the IFN system. We transfected *Ifnar1*$^{-/-}$ MEFs with either dsRNA-*RL* or dsRNA-*GFP* and infected them one day later with a recombinant fully replicative Semliki Forest virus (SFV), a positive-sense (+) ssRNA virus, bearing a *Renilla* luciferase coding sequence

(SFV-Rluc). Notably, transfection with dsRNA-*RL* but not dsRNA-*GFP* protected the cells from SFV-Rluc, leading to a 40- to 70-fold decrease in virus replication, whether measured at the level of *Renilla* luciferase activity or by plaque assay (Fig 6A–C, Appendix Fig S6A). Antiviral activity conferred by dsRNA-*RL* but not dsRNA-*GFP* was similarly observed in *Mavs*$^{-/-}$ MEFs infected with SFV-RLuc (Fig 6C, Appendix Fig S6A). To formally establish that viral restriction was due to dsRNAi, we compared *Ago2*$^{-/-}$

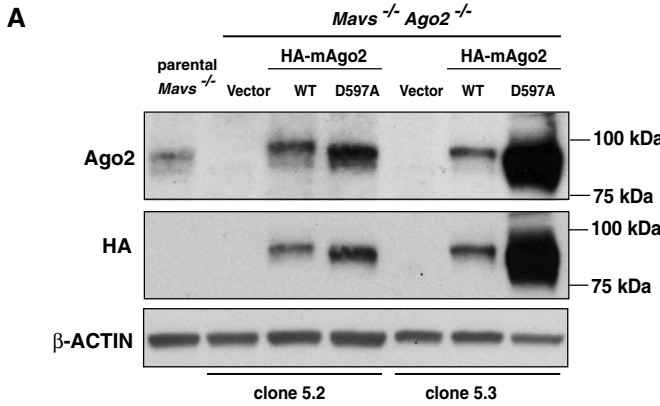

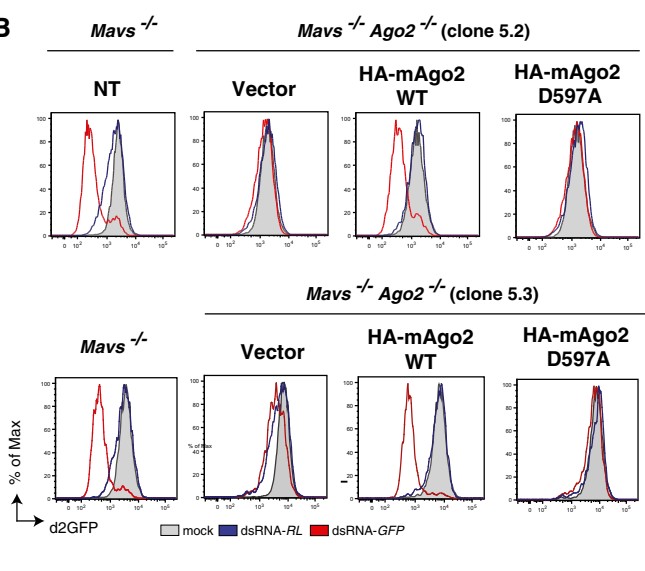

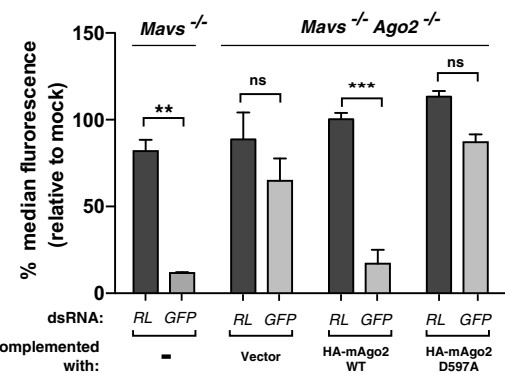

**Figure 5.    The endonucleolytic activity of Ago2 is essential for sequence-specific gene silencing by long dsRNA.**

A   Western blots for Ago2 (top) and HA tag (bottom) of parental *Mavs*$^{-/-}$ MEFs and two *Mavs*$^{-/-}$ *Ago2*$^{-/-}$ clones (clone 5.2 and 5.3; see Fig 4A) stably expressing d2GFP and transduced with an empty vector or a vector encoding HA-tagged wild-type (HA-mAgo2 WT) or catalytic mutant (HA-mAgo2 D597A) versions of mouse Ago2. β-actin served as a loading control. The membrane was first analysed for Ago2 and β-actin, stripped and then analysed for the HA tag.

B   Parental non-transduced (NT) *Mavs*$^{-/-}$ MEFs and complemented *Mavs*$^{-/-}$ *Ago2*$^{-/-}$ MEFs as described in (A) were transfected or not (mock) with the indicated Cy5-labelled dsRNAs, and d2GFP level in Cy5$^+$ cells was monitored by flow cytometry 48 h later.

Data information: All histogram plots are representative of two independent experiments and a sample size of 10,000 cells. Bar graphs depict median fluorescence values normalised to those in mock-transfected samples. Mean values and SD of two independent CRISPR clones transduced with an empty vector or a vector encoding HA-mAgo2 WT or HA-mAgo2 D597A (B) are shown. Statistical analysis was performed using two-way ANOVA with Sidak's multiple comparisons test as post-test for pairwise comparisons. Significant differences with Sidak's multiple comparisons test are shown (ns, not significant; **$P$ < 0.01; ***$P$ < 0.001).

Source data are available online for this figure.

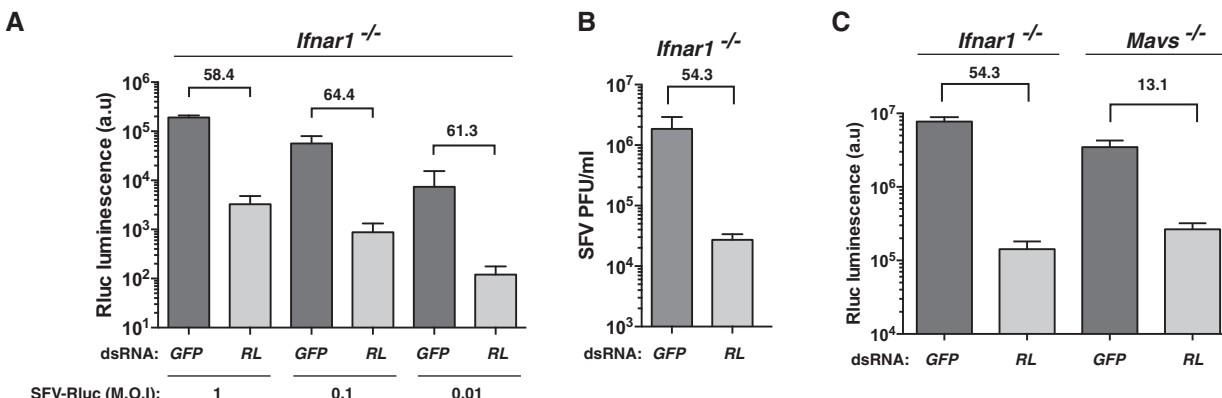

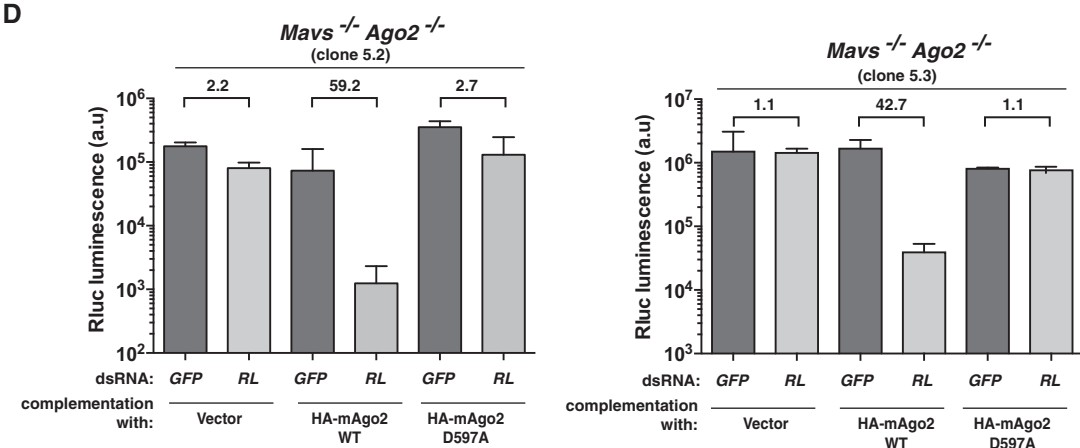

**Figure 6. Long dsRNA vaccinates cells against viruses bearing a homologous sequence.**

A   *Ifnar1*$^{-/-}$ MEFs were transfected with the indicated Cy5-labelled dsRNA and, the next day, they were infected with SFV-Rluc at the indicated multiplicities of infection (MOI). *Renilla* luciferase activity was measured 24 h later.

B   As in (A) but SFV-Rluc virus titre was measured by plaque assay and expressed as plaque-forming units (PFU) per ml of supernatant at 24 h post-infection (MOI = 0.1).

C   As in (A) but with *Ifnar1*$^{-/-}$ or *Mavs*$^{-/-}$ MEFs and a single dose of SFV-Rluc (MOI = 0.1).

D   As in (A) but with two *Mavs*$^{-/-}$ *Ago2*$^{-/-}$ clones (clone 5.2 and 5.3) individually complemented to express wild-type (HA-mAgo2 WT) or a catalytic mutant (HA-mAgo2 D597A) version of mAgo2 or transduced with a control vector (as described in Fig 5). SFV-Rluc was used at an MOI = 0.1.

Data information: Each bar represents mean + SD of biological duplicates, and numbers above each pair of bars depict the fold reduction in viral activity achieved with dsRNA-*RL* vs. dsRNA-*GFP* treatment. Data are representative of four (A), two (B, C) and six (D) independent experiments.

*Mavs*$^{-/-}$ MEFs complemented with wild-type or catalytic mutant HA-mAgo2. Transfection of dsRNA-*RL* conferred antiviral activity against incoming SFV-Rluc in independent *Ago2*$^{-/-}$ *Mavs*$^{-/-}$ MEF clones complemented with wild-type HA-mAgo2 but not in cells transduced with an empty vector or in cells stably expressing the mutant HA-mAgo2 D597A (Fig 6D, Appendix Fig S6B). Thus, dsRNA can vaccinate IFN-deficient cells to provide sequence-specific protection from incoming viruses.

Finally, we tested whether the RNAi pathway is sufficient to restrict virus accumulation in IFN-deficient cells. We first infected *Ago2*$^{-/-}$ *Mavs*$^{-/-}$ and parental *Mavs*$^{-/-}$ MEFs with reovirus (Reo) and Sindbis virus (SINV), a dsRNA virus and a (+)ssRNA virus, respectively. There was wide variation in virus accumulation in different *Ago2*$^{-/-}$ *Mavs*$^{-/-}$ MEFs clones, with no pattern indicative of a consistent difference from the parental *Mavs*$^{-/-}$ MEFs (Appendix Fig S7A and B). To alleviate the effects of

clonal variation, we focused our comparison on single *Ago2*$^{-/-}$ *Mavs*$^{-/-}$ MEFs clones complemented with wild-type vs. catalytic mutant HA-mAgo2. We infected these cells with SFV-Rluc because, as mentioned above, this virus was susceptible to dsRNAi (Fig 6D, Appendix Fig S6B). However, none of the clones displayed any differences in intrinsic resistance to infection (Appendix Fig S7C).

The possible expression of virally encoded VSRs complicates the interpretation of those experiments. Therefore, we also tested the role of RNAi in restriction of a mutant influenza A virus (IAV) lacking NS1 (IAV ΔNS1), a protein previously shown to inhibit the production of long dsRNA-derived siRNAs in plant (Bucher *et al*, 2004; Delgadillo *et al*, 2004; de Vries *et al*, 2009) and human cells (Kennedy *et al*, 2015), as well as prevent antiviral RNAi in *Drosophila* cells (Li *et al*, 2004). However, the replication of IAV ΔNS1, like that of parental wild-type IAV, was similar in *Ago2*$^{-/-}$ *Mavs*$^{-/-}$

MEFs reconstituted to display an intact or a defective RNAi pathway (Appendix Fig S7D). Finally, because IAV is a negative-sense ssRNA virus and may not produce sufficient amounts of dsRNA (Pichlmair *et al*, 2006; Weber *et al*, 2006), we also tested a (+)ssRNA picornavirus, EMCV. As a precaution, we chose an EMCV mutant lacking the L gene, which encodes an IFN antagonist (Hato *et al*, 2007) that could potentially act as a VSR. However, again, we did not see a difference between *Ago2*$^{-/-}$ *Mavs*$^{-/-}$ MEF clones complemented with HA-mAgo2 vs. mutant HA-mAgo2 D597A in resistance to EMCV ΔL (Appendix Fig S7E). Therefore, in our experimental conditions using immortalised MEFs, antiviral RNAi is only patent after vaccination with dsRNA.

## Discussion

Although dsRNAi has been observed in plants and invertebrates, as well as in several undifferentiated mammalian cells, it has so far not been generally documented in differentiated cells (Svoboda *et al*, 2000; Wianny & Zernicka-Goetz, 2000; Billy *et al*, 2001; Yang *et al*, 2001; Paddison *et al*, 2002). Here, we reveal that dsRNAi can be unveiled in somatic cells upon ablation of the IFN response. Notably, dsRNAi is not a rare event restricted to a sub-group of cells but one that can be observed in all cells within the population and that achieves knock-down efficiencies comparable to RNAi induced with siRNA. Finally, activation of this pathway in cells by long dsRNA confers antiviral resistance to subsequent homologous virus infections, indicating that mammalian cells can be "vaccinated" by RNA in a virus-specific manner. Our results establish that somatic cells are fully equipped to mediate functional RNAi from long dsRNA substrates, yet the effects of this activity are generally masked or inhibited by those of the IFN pathway.

How the IFN response prevents experimental observation of dsRNAi is at present unclear. It is conceivable that some ISGs actively suppress dsRNAi. This is likely to occur at the level of Dicer as, in our hands, IFN treatment or the presence or absence of the IFN pathway did not affect siRNA-mediated interference. In contrast, a recent study with human cells reported that activation of the IFN pathway through MAVS led to inhibition of RISC-mediated cleavage and repression activities (Seo *et al*, 2013). Alternatively, the sequence-specific hallmark of dsRNAi might be masked by sequence-unspecific effects resulting from the activity of ISGs, which cause a dsRNA-dependent shutdown in cellular protein synthesis and/or induce cell death (Samuel, 2001; Schneider *et al*, 2014). Irrespective of mechanism, our data suggest that RNAi might be especially important in conditions of limited IFN induction or in cellular niches that possess a reduced IFN response pathway.

In our study, the RNAi pathway was effectively triggered by introduction of long dsRNA and provided sequence-specific resistance to virus infection. Yet, in our experimental system, this resistance was not detected without prior introduction of long dsRNA into the cells. The failure to observe RNAi after virus infection of IFN-deficient cells suggests several non-mutually exclusive possibilities. Many viruses, including vertebrate ones, encode putative VSRs that might obscure RNAi effects on virus accumulation (Ding & Voinnet, 2007; Haasnoot *et al*, 2007; Wu *et al*, 2010). We did not detect an effect of RNAi on resistance to IAV ΔNS1 or EMCV ΔL, yet the role of other putative VSRs warrants further investigation.

Additionally, dsRNAs derived from some viruses (e.g. SFV replication intermediates) might be poor substrates for Dicer due to intrinsic features (e.g. 5′ cap modifications) or due to the fact that they are sequestered inside membrane-bound replication complexes ("viral factories") (Miller & Krijnse-Locker, 2008; den Boon & Ahlquist, 2010). Alternatively, the amplitude and/or kinetics of the dsRNAi response upon virus infection and/or the amount of dsRNA substrate produced by viral replication might not be sufficient to provide effective antiviral protection in the cell types that we studied. In this regard, a possibility to explore is that the magnitude of RNAi is greater in specific cell types, perhaps negatively correlating with IFN responsiveness as discussed above. Interestingly, whereas IFN acts primarily to confer broad-spectrum resistance to viral infection, the RNAi pathway could provide infected cells with an innate mechanism of immune memory and specificity as suggested by our cellular vaccination experiments. Our demonstration of the coexistence of dsRNAi and the IFN response in mammals opens the door to future studies designed to establish the relative contribution of these two forms of antiviral defence to vertebrate protection from viral infection.

## Materials and Methods

### Cells

All cells were mycoplasma negative. E14 mouse embryonic stem cells (mESCs), L929, Vero, BHK-21 cells were obtained from Cancer Research UK Cell Services. HEK 293T cells were a kind gift from Didier Trono (EPFL, Lausanne, Switzerland). L929, Vero and HEK 293T cell lines were cultured in Dulbecco's modified Eagle's medium (DMEM) (Gibco, Life Technologies) supplemented with 10% foetal calf serum (FCS) (Autogen Bioclear UK, Ltd), 2 mM glutamine, 100 U ml$^{-1}$ penicillin, 100 U ml$^{-1}$ streptomycin and were grown at 37°C in 10% $CO_2$. BHK-21 were cultured in Glasgow minimum essential medium (GMEM) (Sigma) containing 10% FCS, 0.2% tryptose phosphate broth (TBP) and 100 U ml$^{-1}$ penicillin, 100 U ml$^{-1}$ streptomycin an grown at 37°C in 10% $CO_2$. E14 mESCs were cultured in GMEM (Sigma), containing 15% embryonic stem cell-qualified FCS (Gibco, Life Technologies), 1,000 U ml$^{-1}$ ESGRO leukaemia inhibitory factor (Millipore), 0.1 mM β-mercaptoethanol (Gibco, Life Technologies), 100 U ml$^{-1}$ penicillin, 100 U ml$^{-1}$ streptomycin, 2 mM glutamine, 1 mM sodium pyruvate (Gibco, Life Technologies) and non-essential amino acids (Gibco, Life Technologies) on a gelatin-coated support in the absence of feeder cells. The culture medium was changed daily. mESCs were grown at 37°C in 8% $CO_2$. MEFs were prepared from 13.5-day *Mavs*$^{-/-}$ or *Ifnar*$^{-/-}$ mouse embryos using standard protocols. They were cultured in DMEM (Gibco, Life Technologies) supplemented to contain 10% FCS (Autogen Bioclear UK, Ltd), 100 U ml$^{-1}$ penicillin, 100 U ml$^{-1}$ streptomycin and 2 mM glutamine. Primary MEFs were then immortalised with simian virus 40 large T antigen as described (Schulz *et al*, 2010).

### Plasmids

MLV-based retroviral vectors were produced using packaging constructs containing Moloney MLV (pCIGPB) *gag-pol* ORF. Lentivirus-based vectors were produced with the packaging construct

pR8.71 and the vector pRRLsin PGK GFP or pRRLsin PGK d2GFP. The env construct for all viral productions was pMD2.G encoding vesicular stomatitis virus G protein. pCIGPB, pR8.71, pRRLsin PGK GFP and pMD2.G were kindly provided by D. Trono. The amino acid coding sequence of mouse Ago2 with a N-terminal epitope derived from influenza virus hemagglutinin (HA) was inserted in pLHCX MLV vector (Clontech) allowing for hygromycin selection of transduced cells. Site-directed mutagenesis of pLHCX-HA-mAGO2 to generate the catalytic mutant version of HA-mAGO2 was performed with the XL QuickChange mutagenesis kit (Stratagene). Specificity of mutagenesis was checked by sequencing. Recombinant SFV-Rluc was generated using the infectious DNA clone pCMV-SFV(3H)Rluc as previously described (Tamberg *et al*, 2007). pCAGGS-Flag-hsDicer was a gift from Phil Sharp (Addgene plasmid # 41584) (Gurtan *et al*, 2012). All primers used to generate pLHCX-HA-mAgo2 WT and pLHCX-HA-mAgo2 D597A are listed in Appendix Table S1.

## Virus infections

Sindbis virus and reovirus strain type 3 Dearing (T3D) were a gift from Ian Kerr and Terence Dermody, respectively. Influenza A/PR/8/34 ΔNS1 was a gift from T. Muster (University of Vienna, Austria). EMCV ΔL mutant was generated from the infectious DNA clone pEC9 EMCV ΔL plasmid (kind gift from Ann C Palmenberg) as described (Deddouche *et al*, 2014). Appropriate doses of virus diluted in 100 μl DMEM without serum were added on cells in a 24-well plate and left for 2 h at 37°C. The medium was then replaced with 500 μl of DMEM supplemented with 10% FCS, 2 mM gluta-mine, 100 U ml$^{-1}$ penicillin, 100 U ml$^{-1}$ streptomycin, and cells were then incubated at 37°C. For measurements of viral load by qRT–PCR, cells were washed with PBS, trypsinised and collected by centrifugation. After a subsequent wash with PBS, cells were lysed and RNA extracted using RNeasy Mini Kit (Qiagen). For measurements of infectious particles accumulation, cell supernatants were collected, clarified by centrifugation at 1,000 *g* for 15 min, aliquoted and frozen at −80°C for subsequent analysis by plaque assay.

## Preparation of long dsRNA

PCR fragments corresponding to the first 200 nt of the GFP coding sequence were amplified from pRRL-PGK-GFP using primers additionally containing a T7 promoter sequence, with one PCR fragment allowing the production of (+)-sense RNA GFP and the other used to generate the (−)-sense RNA. Similarly, two PCR fragments corresponding to the first 200 nt of *Renilla* luciferase were amplified from pRL-TK plasmid (promega). Primers used to generate the PCR products are listed in Appendix Table S1. The PCR fragments were purified using QIAquick PCR purification kit (Qiagen), and *in vitro* transcription (IVT) with T7 RNA polymerase (T7 MEGAscript kit, Ambion) was performed overnight at 37°C. To label the IVT RNA, 1/10$^{th}$ Cy5-CTP (Amersham CyDye Fluorescent Nucleotides Cy5-CTP, GE Healthcare Life sciences) was included in the IVT reaction. The next day, IVT RNA was digested with Turbo DNase for 30 min at 37°C and unincorporated nucleotides were removed by purifying the IVT RNA on an Illustra MicroSpin G-25 column (GE Healthcare Life), followed by phenol-chloroform extraction and an additional purification on a G-25 column. The correct sizes of the IVT RNAs were verified on a TBE 2% agarose gel. DsRNA was then generated

by mixing an equal volume of (+)-sense IVT RNA and (−)-sense IVT RNA in annealing buffer (5 mM Tris pH 7.5, 25 mM NaCl, 0.5 mM EDTA), incubation at 65°C for 10 min followed by slow cool down to RT. To remove single-stranded RNA molecules, the mixture was treated with RNase I, a single-stranded RNA-specific RNase for 10 min. dsRNA was then purified by phenol-chloroform extraction, quantified by Nanodrop and separated on a TBE 2% agarose gel to verify size and integrity. RNA was visualised by ethidium bromide staining and in-gel fluorescence using IMAGEQUANT LAS 4000 (GE Healthcare).

## Viral vector production

All vector production was performed by CaPO$_4$-mediated transient co-transfection of the retroviral vector, *gag-pol* and *env* encoding constructs. Briefly, subconfluent HEK 293T cells were co-transfected with 21.5 μg of vector, 14.6 μg packaging constructs and 7.9 μg env constructs in a 15-cm plate. Cells were washed 16 h post-transfection, and supernatants were harvested 24 h later. Recombinant retroviral vectors containing supernatants were centrifuged, filtered and concentrated by ultracentrifugation.

## Engineering of stable cell lines

To generate GFP or d2GFP-expressing lines, lentiviral vectors encoding GFP or d2GFP were produced using pRRLsin PGK GFP or pRRLsin PGK d2GFP plasmids as described above. Concentrated lentiviral particles were added to 5 × 10$^4$ target cells. 72 h post-transduction, cells were FACSsorted (on a BD FACSAria or a BDFACSAria Fusion) for GFP-positive cells and expanded. MLV-based retroviral vectors encoding wild-type or mutant version of mouse Ago2 were produced using the pLHCX-derived plasmids, and concentrated retroviral vectors were added to 5 × 10$^4$ target cells. Two days post-transduction, cells were expanded and transduced cells were selected by adding hygromycin B (Sigma) at a concentration of 200 μg ml$^{-1}$.

## Dicer knock-down cells

Five MISSION pLK0.1-puro Dicer shRNA (Sigma) and MISSION pLK0.1-puro non-target shRNA control plasmids (Sigma) were used together with lentiviral packaging and envelope constructs to produce individual lentiviral vectors particles as described above. *Mavs*$^{-/-}$ stably expressing d2GFP were transduced with lentiviral vectors. Knock-down efficiency was assessed by immunoblot, and two lines stably expressing shRNA against Dicer and cells expressing non-target control shRNA were used for experiments.

## Production of recombinant SFV-Rluc

BHK-21 cells at 80% confluency in a single well of a six-well plate were transfected by Lipofectamine 2000 (Invitrogen) with 1 μg of infectious DNA plasmid pCMV-SFV(3H)Rluc. The transfection mix was first incubated for 20 min at RT, cells were washed once with OptiMEM, and the transfection mixture was then directly added on the cells. After 3 h of incubation at 37°C, the mixture was removed and infection medium (GMEM, 0.2% BSA, 20 mM Hepes) was added. Cells were incubated for 24 h at 37°C until the first signs of a cytopathology. The supernatant was then harvested and clarified by

centrifugation (1,000 *g*, Eppendorf 5810, swing rotor) for 15 min at 4°C and by filtration (0.4 µm filters) to remove remaining cells and cell debris. Glycerol was added to a final concentration of 5%, and viral stocks were aliquoted and stored at −80°C. Stocks were titrated by plaque assay on BHK-21 cells.

### CRISPR-mediated genome engineering

*Mavs*$^{-/-}$ *Ago2*$^{-/-}$ and *Mavs*$^{-/-}$ *Ago1*$^{-/-}$ cells stably expressing GFP (Appendix Fig S4) or d2GFP (Fig 4) were generated by CRISPR-Cas9-mediated genome engineering using the CRISPR design tool provided by the Zhang laboratory (http://www.genome-engineering.org). A target sequence in the second exon (GTGGTGCCGAAGTCCGGCCGAGG, sgRNA 1, Fig 4, Appendix Fig S4) or in the fourth exon (GATCGTCTCGAAGGGGACGCTGG, sgRNA 2, Appendix Fig S4) of murine Ago2 was chosen, and a target sequence in the second exon (GATTGACGTTTACCATTACGAGG, Appendix Fig S4) of murine Ago1 was selected. The appropriate oligonucleotides were cloned into the BbsI site of pSpCas9(BB)-2A-puro plasmid obtained from the laboratory of Feng Zhang via Addgene (https://www.addgene.org, plasmid 48139) according to the cloning protocol provided by the laboratory (http://www.genome-engineering.org). Following puromycin selection (48 h) and single-cell cloning, colonies were screened for correct targeting of Ago2 by measuring the efficiency of siRNA-mediated knock-down of GFP. Ago2 and Ago1 deficiency was confirmed by immunoblot. All primers used for CRISPR/Cas9-mediated genome engineering in this study are listed in Appendix Table S1.

### RT–PCR

Total RNA was extracted using RNeasy Mini Kit columns with DNase treatment according to the manufacturer's instructions (Qiagen). Five hundred ng of RNA was reverse transcribed using random hexamers (Thermo Scientific) and SuperScript II Reverse Transcriptase (Thermo Fisher Scientific). cDNA was then diluted five times in nuclease-free water and analysed for gene expression by qPCR using TaqMan Gene expression Assays (for GFP and *Gapdh*, Applied Biosystems) or SYBR green (Applied Biosystems). Reactions were carried out using ABI 7500 Fast machines (Applied Biosystems). Relative expression values were calculated using the ΔΔ*C*t method. *C*t values were normalised to *Gapdh* or *Hprt* housekeeping gene. All primers used for qRT–PCR using SYBR green are listed in Appendix Table S1.

### DsRNA or siRNA transfection and flow cytometry

$4 \times 10^4$ cells per well were seeded on a 24-well plate. One day later, 200 ng of Cy5-labelled dsRNA-*RL* or Cy5-dsRNA-*GFP* or 15 pmol siRNA-GFP (Silencer GFP siRNA, Ambion) or 15 pmol siRNA control (Silencer negative control siRNA # 1, Ambion) or transfection reagent alone (mock) was transfected into cells using Lipofectamine 2000 (Invitrogen). To test the effect of IFN on dsRNA/siRNA-mediated gene silencing, 200 U of recombinant IFN A/D (PBL Assay Science) was added at the time of seeding the cells. To block the IFN receptor, an anti-IFNAR antibody (LEAF purified anti-mouse IFNAR-1 antibody, Biolegend) was included at 10 µg ml$^{-1}$. An isotype-matched irrelevant specificity antibody (LEAF purified

mouse IgG1, k isotype control antibody, Biolegend) was used as a control. For analysis, cells were recovered by trypsinisation, washed in PBS and resuspended in FACS buffer (PBS, 1% FCS, 5 mM EDTA) containing the live/dead cell discriminator dye DAPI, as well as a fixed amount of fluorescent reference beads (Coulter CC Size standard L10, Beckman Coulter) in order to allow for flow cytometry-based cell counting. Flow cytometric analyses were performed with an LSRFortessa (BD Biosciences) acquiring a sample size of 10,000 cells to meet statistical robustness. Data were analysed using FlowJo (Tree Star).

### Small RNA Northern blot

$2 \times 10^6$ cells were seeded on a 10-cm dish. One day later, 4.5 µg of dsRNA-*RL* or dsRNA-*GFP* or transfection reagent alone (mock) was transfected into cells using Lipofectamine 2000 (Invitrogen). 24 h post-transfection, total RNAs were extracted and purified using TRIzol Reagent (Invitrogen) according to manufacturer's instructions and resuspended in 50% formamide. The yield was determined using a spectrophotometer and 25 µg of total RNAs as well as 24 ng of miRNA marker (New England Biolabs) and 2 µl of *in vitro* dicing reactions (see section below) were resolved on denaturing 17.5% polyacrylamide/urea gels, transferred on a Hybond™-NX membrane (GE Healthcare) and chemically cross-linked using 1-ethyl-3-(3-dimethylaminopropyl) carbodiimide (EDC) as previously described (Pall & Hamilton, 2008). Perfect-Hyb buffer (Sigma) was used for the hybridisation step. DNA oligonucleotides complementary to U6 or to the miRNA marker (New England Biolabs) were 5′end-labelled with γ-$^{32}$P-ATP (PerkinElmer) using T4 polynucleotide kinase (Thermo Scientific). The probe used to detect GFP or *Renilla* luciferase siRNAs was made by random priming in the presence of α-$^{32}$P-dCTP (PerkinElmer) using the Prime-a-gene kit (Promega). The templates used for these random priming reactions were PCR fragments corresponding to the first 200 nt of the GFP coding sequence and *Renilla* luciferase, respectively. The PCR fragments were purified using QIAquick PCR purification kit (Qiagen). Membranes were stripped with boiling 0.1% SDS solution twice. Probes used for Northern blot are listed in Appendix Table S1.

### Purification of recombinant human Dicer and *in vitro* dicing assay

pCAGGS-Flag-hsDicer (Addgene plasmid # 41584) (Gurtan *et al*, 2012) was transfected into 293T cells using Lipofectamine 2000 according to the manufacturer's instructions. Twenty-eight hours post-transfection, cells were lysed in lysis buffer (30 mM Tris pH 6.8, 50 mM NaCl, 3 mM MgCl$_2$, 5% glycerol, 0.4% NP-40 and protease inhibitors (Roche)) and the lysate was cleared by centrifugation. FLAG-Dicer was immunoprecipitated using anti-FLAG M2 beads (Sigma), washed 5 × in lysis buffer and 1 × in lysis buffer without NP-40. FLAG-Dicer was eluted from the beads using 0.5 mg ml$^{-1}$ 3× FLAG Peptide (Sigma) and lysis buffer without NP-40. The purity and concentration of FLAG-Dicer was analysed by SDS–PAGE and Coomassie staining, using BSA as a standard. 500 nM FLAG-Dicer was incubated with 50 nM dsRNA in dicing buffer (30 mm Tris pH 6.8, 50 mM NaCl, 3 mM MgCl$_2$, 5% glycerol, 4 mM DTT, RNAsin) for 1 h at 37°C. Reactions were purified by

phenol/chloroform extraction, resuspended in 50% formamide and analysed by small RNA Northern blot.

## Protein analysis

Cell lysates were prepared in RIPA buffer (PBS with 1% NP-40, 0.5% sodium deoxycholate and 0.1% SDS) supplemented with protease inhibitor cocktail (Complete®, Roche). Protein content was quantified using the Pierce BCA Protein Assay Kit, and equal amounts of protein were resolved on a Tris–glycine SDS–polyacrylamide gel and transferred by electroblotting onto Inmobilon®-P PVDF membranes (Millipore) and incubated sequentially with the relevant primary antibodies and HRP-conjugated secondary antibody in PBS with 0.1% Tween-20 and 5% non-fat dried milk. Luminata Crescendo Western HRP substrate (Millipore) was used for detection. For re-probing, membrane was stripped using Restore Western blot Stripping Buffer (Thermo Scientific) according to manufacturer's instructions. Stripping efficiency was verified by treating the stripped membrane with Luminata Crescendo HRP substrate and confirming the absence of any signal following 1-h exposure. The following antibodies were used: HA-HRP (clone 3F10, Roche), Ago2 (C34C6, Cell Signaling Technology), Ago1 (D84610, Cell Signaling Technology), Dicer, MAVS (Cell Signaling Technology), GFP-HRP (D5.1, Cell Signaling Technology), β-actin-HRP (13E5) and p97 (Fitzgerald Industries).

## Plaque assay

BHK-21 or Vero cells were seeded at $4 \times 10^5$ cells per well in a 12-well plate. The next day, 220 µl of 10-fold serial dilutions of virus-containing supernatant (diluted in DMEM without FCS) was added to the cells and left for 2 h at 37°C. The infection mix was then aspirated and DMEM containing 2% FCS and 0.8% SeaPlaque Agarose (Lonza) was added and left for 10 min at room temperature to allow the agarose to solidify. Cells were then cultured at 37°C in 10% $CO_2$ for 2 days after which plaques were stained for 2 h at 37°C by adding 0.1 volume of MTT solution (5 mg ml$^{-1}$ in PBS; Sigma).

## Assay to monitor the antiviral activity of dsRNAi

Cells were seeded and transfected with long dsRNA as described above. One day after transfection, the medium was removed and the required dose of SFV-Rluc (diluted in serum-free DMEM) was added and left for 2 h at 37°C. The infection mix was then removed, and complete medium (DMEM containing 10% FCS and antibiotics) was added on the cells. Twenty-four hours post-infection, the medium was recovered for analysis of viral concentration by plaque assay and cells were lysed in 1× Passive Lysis buffer (Promega). Luciferase activity was analysed in cell lysates using the Dual-Luciferase Reporter Assay System (Promega) and measured on an Envision 2102 Multilabel Reader (PerkinElmer).

## Statistical analysis

Statistical analysis was performed using GraphPad Prism 6. A *P*-value < 0.05 was considered statistically significant.

**Expanded View** for this article is available online.

## Acknowledgements

We thank Helen Rowe, Jean-Luc Imler and members of the Immunobiology Laboratory for helpful discussions and suggestions. We also thank Oliver Gordon for advice on statistical analysis and the Crick Equipment Park as well as the Flow Cytometry Facility for technical assistance. This work was supported by The Francis Crick Institute, which receives its core funding from Cancer Research UK (FC001136), the UK Medical Research Council (FC001136) and the Wellcome Trust (FC001136). P.V.M. was supported by an Advanced Postdoc Mobility fellowship from the Swiss National Science Foundation and by a Marie-Curie actions Intra-European fellowship. A.G.V. was supported by an EMBO Long-Term Fellowship and a Rubicon Fellowship from the Netherlands Organization for Scientific Research.

## Author contributions

PVM and CRS designed experiments, analysed data and wrote the manuscript. PVM conducted experiments with assistance from AGV, SD and NCR. AM provided key reagents. CRS supervised the project.

## Conflict of interest

The authors declare that they have no conflict of interest.

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
