## [Review Process File · The EMBO Journal]

Manuscript EMBO-2016-95086

Inactivation of the type I interferon pathway reveals long double stranded RNA-mediated RNA interference in mammalian cells

Pierre V Maillard, Annemarthe G Van der Veen, Safia Deddouche-Grass, Neil C Rogers, Andres Merits and Caetano Reis e Sousa

Corresponding author: Caetano Reis e Sousa & Pierre Maillard, The Francis Crick Institute

Review timeline:

Submission date:	22 June 2016
Editorial Decision:	18 July 2016
Revision received:	02 September 2016
Acceptance:	12 September 2016
Accepted:	12 September 2016

Editor: Karin Dumstrei

Transaction Report:

1st Editorial Decision

18 July 2016

Thanks for submitting your manuscript to The EMBO Journal. I am sorry for the delay in getting back to you, but I have now received comments from the two referees on your manuscript.

As you can see below, both referees find the analysis well done and the data conclusive. A concern raised by referee #1 is that we gain too limited mechanistic insight into how the IFN suppresses the dsRNAi pathway. However, referee #2 also highlights the importance of the findings. While further mechanistic insight into the process would of course be nice I also find that we don't such insight at this stage.

Given the comments from the referees, I would like to ask you to submit a suitably revised manuscript. As mentioned above we don't need any further mechanistic insight, but I would be keen to here you response to referee #1 point 2. The statistical analysis also has to be sorted out. Referee #2 raises a number of constructive points that I am sure that you must have considered and are in a position to respond to. I am available to discuss the specifics further.

When preparing your letter of response to the referees' comments, please bear in mind that this will form part of the Review Process File, and will therefore be available online to the community. For more details on our Transparent Editorial Process, please visit our website: http://emboj.emboPress.org/about#Transparent_Process

We generally allow three months as standard revision time. As a matter of policy, competing manuscripts published during this period will not negatively impact on our assessment of the

conceptual advance presented by your study. However, we request that you contact the editor as soon as possible upon publication of any related work, to discuss how to proceed. Should you foresee a problem in meeting this three-month deadline, please let us know in advance and we may be able to grant an extension.

Thank you for the opportunity to consider your work for publication. I look forward to your revision.

REFEREE REPORTS

Referee #1:

In this work Maillard et al have examined the role of dsRNAi in antiviral defense in mammalian cells. The work expands on an article from 2013 by Maillard (Science 342:235) demonstrating that RNAi is active as an antiviral mechanism in mammalian cells. The idea behind the work is interesting, and the authors try to prove that lack of type I IFN uncovers a role for dsRNAi in antiviral defense. Although the data presented in the present work is well designed, and the conclusions drawn by the authors are generally supported by the data, the work appears underdeveloped.

1. The results are very descriptive, and lack mechanistic explanation for how IFN/ISGs suppress sequence-specific responses to long dsRNAs.
2. Critically, data are missing on the role/impact of endogenous dsRNAi in antiviral defense in *Ifnar*^{-/-} cells/mice.
3. Throughout, the lack of statistical analysis of the data weakens the conclusions.

Referee #2:

Review of EMBOJ-2016-95086

Maillard et al examine the presence of a functional RNAi pathway in differentiated mammalian cells after inactivating perhaps more dominant innate immune pathways regulated by PRRs or IFN. The authors use long dsRNA as a probe and show dsRNA specificity for repression of GFP expression using a flow cytometry-based assay. The results and controls support the existence of a functional RNAi pathway and cells "immunized" with dsRNA can inhibit replication of an SFV derivative with the target sequence. The paper is well written and the conclusions tempered where appropriate. What is not clear is whether or not this pathway actually plays an important role in mammalian antiviral protection since it does not appear to be sufficiently strong unless cells are first primed with dsRNA. Given the rather polarized views regarding functional RNAi in mammalian cells, a few additional experiments might be in order.

Major points:

1. Can the authors actually show that target-specific small RNAs of the appropriate size are present in the dsRNA treated cells?
2. Although when taken together the control experiments support the authors' thesis, it would be reassuring and bolster their argument for sequence specificity to show that a different, non-targeted fluorescent reporter (say RFP or a recoded GFP to ablate the target homology) with otherwise the same mRNA structure does not exhibit decreased expression by the dsRNA that inhibits GFP expression. Off target effects of transfected nucleic acids are always a concern and a non-targeting dsRNA of markedly different sequence may not be a sufficiently rigorous control.

Minor points:

1. For most of the experiments, silencing induced by dsRNA was not restricted to a subset and the entire transfected population shifted towards the left, with one peak. However, some of the data

(Fig.3B, Fig.4D), there are two distinct peaks for dsRNA-GFP transfection (one shifted to low expression, the other overlapped with Mock). Does this imply that the efficacy of silencing can be affected by other factors?

2. For the Fig. 5A western blot, it might be desirable to add a lane with parental Mavs^{-/-} cells probed with Ago2 antibody, to show the levels of restored expression of Ago2. Theoretically, the level of Ago2 should be critical for the shift.

3. There are several possibilities for the failure to observe RNAi after viral infection of IFN-deficient cells: (1) VSRs obscure RNAi effects; (2) dsRNAi is not robust enough to provide effective antiviral effects. Probably beyond the scope of this initial report, but finding a natural context where the IFN response is suppressed and this RNAi type of response takes over would be ideal. But this might be very difficult to find if selection pressure for antiviral protection has shifted to the IFN and protein based innate immune responses (especially those focused on dsRNA sensing). Thus while the RNAi factors have been co-opted in vertebrates for other purposes, it might not be too surprising that there is some low level "vestigial" activity. A prediction/question that could be tested: If you test VSR deficient viruses, would you expect to see less efficient replication in parental Mavs^{-/-} cells, compared to Ago2^{-/-} Mavs^{-/-} cells?

4. There are many "representative" data panels, rather than showing the average, error bars (standard deviation) and statistical measures of significance. Figure 1G, for example, lacks error bars. Given that the effects are rather subtle in some of the experiments, it is important to give readers (and reviewers) an idea of how much variability is observed between replicate experiments.

1st Revision - authors' response

02 September 2016

Referee #1

In this work Maillard et al have examined the role of dsRNAi in antiviral defense in mammalian cells. The work expands on an article from 2013 by Maillard (Science 342:235) demonstrating that RNAi is active as an antiviral mechanism in mammalian cells. The idea behind the work is interesting, and the authors try to prove that lack of type I IFN uncovers a role for dsRNAi in antiviral defense. Although the data presented in the present work is well designed, and the conclusions drawn by the authors are generally supported by the data, the work appears underdeveloped.

1. The results are very descriptive, and lack mechanistic explanation for how IFN/ISGs suppress sequence-specific responses to long dsRNAs.

We agree with the reviewer that our study does not explain how ISGs suppress long dsRNAi in mammalian cells. However, the mere existence of the process has been so controversial that we felt that, in a first report, it would be most important to thoroughly document it and investigate its possible functional consequences. We hope to investigate the mechanistic underpinnings of our observations in subsequent studies.

2. Critically, data are missing on the role/impact of endogenous dsRNAi in antiviral defense in Ifnar^{-/-} cells/mice.

We thank the reviewer for suggesting that we expand on this aspect, which was missing from our original submission. We have performed a series of experiments that we have now included as part of Appendix Figure S7. We have performed infections with reovirus, Sindbis virus, influenza A virus and encephalomyocarditis and have failed to find evidence for an impact of dsRNAi on virus accumulation. The possible implications of these data are mentioned in the discussion of our manuscript.

3. Throughout, the lack of statistical analysis of the data weakens the conclusions.

We have previously presented flow cytometry data from single representative experiments. This is customary in FACS analysis because such data are not easily amenable to pooling between experiments. For example, median fluorescence intensity values are not absolute and depend on the machine used, its settings and, even if a single machine and identical settings are used, the extent of laser drift between experiments. Nevertheless, we have now been able to pool data from

independent experiments by normalising in each experiment the median fluorescence intensity values from each sample to those of the mock control. We now show bar graphs depicting these pooled data with standard deviations and statistical analysis carried out using two-way ANOVA or unpaired *t*-test (if only 2 conditions are compared). For Figure 6, exact virus titres obtained after infection can vary from one experiment to the next and pooling generates large errors, which are absent when replicate samples within a single experiment are compared. Therefore, we show data for a representative experiment for clarity, yet we added to Appendix Figure S6 a summary of all the data from all independent experiments. These all show the same effect and confirm the robustness of our observations.

Referee #2:

Maillard et al examine the presence of a functional RNAi pathway in differentiated mammalian cells after inactivating perhaps more dominant innate immune pathways regulated by PRRs or IFN. The authors use long dsRNA as a probe and show dsRNA specificity for repression of GFP expression using a flow cytometry-based assay. The results and controls support the existence of a functional RNAi pathway and cells "immunized" with dsRNA can inhibit replication of an SFV derivative with the target sequence. The paper is well written and the conclusions tempered where appropriate. What is not clear is whether or not this pathway actually plays an important role in mammalian antiviral protection since it does not appear to be sufficiently strong unless cells are first primed with dsRNA. Given the rather polarized views regarding functional RNAi in mammalian cells, a few additional experiments might be in order.

Major points:

1. Can the authors actually show that target-specific small RNAs of the appropriate size are present in the dsRNA treated cells?

We now include as part of Fig 4A a Northern blot of total RNA from *Ifnar1*^{-/-} MEFs transfected with dsRNA-RL or dsRNA-GFP. Using probes specific for either dsRNA-GFP or dsRNA-RL, we could detect 22-nt cleavage products that co-migrated with Dicer-dependent siRNAs generated *in vitro*. We thank the reviewer for encouraging us to perform this experiment, the results of which have strengthened our manuscript.

2. Although when taken together the control experiments support the authors' thesis, it would be reassuring and bolster their argument for sequence specificity to show that a different, non-targeted fluorescent reporter (say RFP or a recoded GFP to ablate the target homology) with otherwise the same mRNA structure does not exhibit decreased expression by the dsRNA that inhibits GFP expression. Off target effects of transfected nucleic acids are always a concern and a non-targeting dsRNA of markedly different sequence may not be a sufficiently rigorous control.

The referee raises the concern that the effect that we see on GFP expression using dsRNA-GFP might be caused by off-target effects restricted to dsRNA-GFP but not with dsRNA-RL. Yet, we could observe that the dsRNA-RL is also processed into siRNAs (see section above and Fig 4A) and, more importantly, the same dsRNA-RL is used against SFV-RLuc in the experiments in Fig 6, in which dsRNA targeting GFP is now used as a control. We observe a sequence-specific antiviral activity against SFV-RLuc provided by the cognate dsRNA-RL compared to dsRNA-GFP. This criss-cross effect is, in our view, the best measure of specificity as it demonstrates clearly that each dsRNA used in the paper has RNAi activity that specifically impacts only the expression of its cognate target.

Minor points:

1. For most of the experiments, silencing induced by dsRNA was not restricted to a subset and the entire transfected population shifted towards the left, with one peak. However, some of the data (Fig.3B, Fig.4D), there are two distinct peaks for dsRNA-GFP transfection (one shifted to low expression, the other overlapped with Mock). Does this imply that the efficacy of silencing can be affected by other factors?

The efficiency of silencing correlates positively with the efficiency of transfection with Cy5 labeled dsRNA (Cy5-dsRNA). For most experiments, the level of transfection was such that the majority of the cells became Cy5^{high} when analysed by flow cytometry. Yet, in some experiments, transfection was less efficient resulting in cells that were Cy5^{low} to Cy5^{high}. Given the reviewer's comment we reanalyzed the data from Fig. 3B using a more stringent gate to select more specifically the Cy5^{high} cells that had been transfected most efficiently. We replaced the plots of Fig. 3B with the analysis obtained with this more stringent gating strategy. The reviewer also mentioned the plots of Fig 4 D (now part of Fig 4 C). For those analyses, a more stringent gating was not appropriate as this resulted in a very low number of cells within the gate. We therefore left the panels of Fig.4 C unchanged but added a note in the figure legend to provide this information to the reader.

2. For the Fig. 5A western blot, it might be desirable to add a lane with parental *Mavs*^{-/-} cells probed with Ago2 antibody, to show the levels of restored expression of Ago2. Theoretically, the level of Ago2 should be critical for the shift.

We have repeated the Western blot including, as requested, the parental *Mavs*^{-/-} cells and incorporated it into Fig 5A. We probed the membrane first with Ago2 antibody and then stripped it off and probed it with an anti-HA antibody. We found that the level of endogenous Ago2 in parental *Mavs*^{-/-} cells is similar to the level of Ago2 detected in the 2 *Mavs*^{-/-} *Ago2*^{-/-} clones complemented with HA-mAgo2 WT. The sequence-specific gene silencing is observed in these 3 cell lines but not in cells that do not express Ago2 (*Mavs*^{-/-} *Ago2*^{-/-} transduced with empty vector). The sequence-specific effect is also abolished in cells that express a catalytic mutant of Ago2 (HA-mAgo2 D597A) either at similar (lane 4) or much higher levels (lane 7) than endogenous Ago2 parental *Mavs*^{-/-} cells.

3. There are several possibilities for the failure to observe RNAi after viral infection of IFN-deficient cells: (1) VSRs obscure RNAi effects; (2) dsRNAi is not robust enough to provide effective antiviral effects. Probably beyond the scope of this initial report, but finding a natural context where the IFN response is suppressed and this RNAi type of response takes over would be ideal. But this might be very difficult to find if selection pressure for antiviral protection has shifted to the IFN and protein based innate immune responses (especially those focused on dsRNA sensing). Thus while the RNAi factors have been co-opted in vertebrates for other purposes, it might not be too surprising that there is some low level "vestigial" activity. A prediction/question that could be tested: If you test VSR deficient viruses, would you expect to see less efficient replication in parental *Mavs*^{-/-} cells, compared to *Ago2*^{-/-} *Mavs*^{-/-} cells?

As suggested by the reviewer, an impact of RNAi on virus infection might be more easily revealed using VSR-deficient viruses. We therefore tested influenza virus ΔNS1 because NS1 was shown to have VSR activity in *Drosophila* cells¹ and inhibit production of siRNA from dsRNA substrate in plants²⁻⁴ and human cells⁵. However, we did not detect a difference in the replication of influenza ΔNS1 (or parental virus) in cells displaying or not a functional RNAi pathway. Influenza virus is negatively stranded and may not produce sufficient amounts of dsRNA^{6,7}. Therefore, we also tested a positively-stranded picornavirus, EMCV, lacking the L protein (which might constitute another possible VSR on the basis that it inhibits IFN responses). Again, we did not see an effect of antiviral RNAi with EMCV ΔL. We have included these data, together with data from other virus infections experiments, as part of Appendix Figure 7 and discuss their implications in the text.

4. There are many "representative" data panels, rather than showing the average, error bars (standard deviation) and statistical measures of significance. Figure 1G, for example, lacks error bars. Given that the effects are rather subtle in some of the experiments, it is important to give readers (and reviewers) an idea of how much variability is observed between replicate experiments.

Please see our answer to comment 3 of reviewer 1.

References.

1. Li, W. X. *et al.* Interferon antagonist proteins of influenza and vaccinia viruses are suppressors of RNA silencing. *Proc Natl Acad Sci U S A* **101**, 1350–1355 (2004).
2. Delgadillo, M. O., Sáenz, P., Salvador, B., García, J. A. & Simón-Mateo, C. Human influenza virus NS1 protein enhances viral pathogenicity and acts as an RNA silencing suppressor in plants. *J Gen Virol* **85**, 993–999 (2004).

3. Bucher, E., Hemmes, H., de Haan, P., Goldbach, R. & Prins, M. The influenza A virus NS1 protein binds small interfering RNAs and suppresses RNA silencing in plants. *J Gen Virol* **85**, 983–991 (2004).
4. de Vries, W., Haasnoot, J., Fouchier, R., de Haan, P. & Berkhout, B. Differential RNA silencing suppression activity of NS1 proteins from different influenza A virus strains. *J Gen Virol* **90**, 1916–1922 (2009).
5. Kennedy, E. M. *et al.* Production of functional small interfering RNAs by an amino-terminal deletion mutant of human Dicer. *Proceedings of the National Academy of Sciences* (2015). doi:10.1073/pnas.1513421112
6. Weber, F., Wagner, V., Rasmussen, S. B., Hartmann, R. & Paludan, S. R. Double-stranded RNA is produced by positive-strand RNA viruses and DNA viruses but not in detectable amounts by negative-strand RNA viruses. *J. Virol.* **80**, 5059–5064 (2006).
7. Pichlmair, A. *et al.* Activation of MDA5 requires higher-order RNA structures generated during virus infection. *J. Virol.* **83**, 10761–10769 (2009).

Accepted

12 September 2016

Thank you for submitting your revised manuscript to The EMBO Journal. Your study has now been seen by referee #2 and the comments are provided below. As you can see the referee appreciates the introduced changes and balanced discussion. I am therefore very pleased to accept the manuscript for publication.

REFEREE REPORT

Referee #2:

The authors have adequately addressed our main concerns. Kudos for doing the VSR deficient virus experiment. Not seeing any difference on the surface sinks the argument that this is an ancient "backup" system. However, I'd be hard pressed to fully believe this negative result given the super artificial (but state of the art) cell culture context used. Their language on why they could be missing antiviral dsRNAi reads appropriately.

Bottom line: This is a result awaiting functional impact, but well reasoned and controlled. It opens up future work to look for a context where this pathway makes a difference.

Corresponding Author Name: Caetano Reis e Sousa

Journal Submitted to: The EMBO Journal

Manuscript Number: EMBOJ-2016-95086